# Contrastive Learning of 3D Shape Descriptor with Dynamic Adversarial Views

## Abstract

View-based deep learning models have shown the capability to learn 3D shape descriptors with superior performance on 3D shape recognition, classification, and retrieval. Most popular techniques often leverage class labels to train deep neural networks under supervision to learn to extract 3D deep representation by aggregating information from a static and pre-selected set of different views used for all shapes. Those approaches, however, often face challenges posed by the requirement of a large amount of annotated training data and the lack of a mechanism for the dynamic mining of shape-instance-dependent views towards the learning of more informative 3D shape representation. This paper addresses those two challenging issues by introducing the concept of adversarial views and developing a new mechanism to generate 2D views for the adversarial training of a self-supervised contrastive model for 3D shape descriptor, denoted as CoLAV. In particular, compared to the recent advances in multi-view approaches, our proposed CoLAV gains advantages by leveraging the contrastive learning techniques for self-supervised learning of 3D shape representations without the need for labeled data. In addition, CoLAV introduces a novel mechanism for the dynamic generation of shape-instance-dependent adversarial views as positive pairs to adversarially train robust contrastive learning models towards the learning of more informative 3D shape representation. Comprehensive experimental results on 3D shape classification demonstrate that the 3D shape descriptor learned by CoLAV exhibits superior performance for 3D shape recognition over other existing view-based descriptor learning techniques, even though CoLAV is completely self-trained with unlabeled 3D datasets (e.g., ModelNet40).

## 1 Introduction

The impressive performance of Deep Neural Networks (DNNs) for 2D visual feature learning naturally motivates 3D computer vision researchers to develop deep learning models for 3D representation for downstream 3D tasks (e.g., shape classification (Qi et al., 2017b), shape correspondence (Li et al., 2019), shape retrieval (Han et al., 2018), shape registration (Miao et al., 2016). To this end, researchers developed the first family of 3D deep learning models that operate directly on 3D raw data represented by point clouds (Qi et al., 2017b; Shi et al., 2020), triangulated meshes (Hanocka et al., 2019), or 3D regular voxels (Choy et al., 2019). The recent advancement of approaches in this category has significantly contributed to the shape classification, correspondence, segmentation, registration, and detection. The second important family of 3D deep learning models starts with rendering multiple 2D projections of a 3D object from a static and pre-selected set of different views, followed by learning the compact 3D representation via fusing the deep feature maps of these 2D views. As they can straightforwardly leverage the latest developments of convolutional neural networks, the view-based methods have shown a very attractive capability to learn 3D shape descriptors with superior performance on 3D shape recognition, classification, and retrieval (Li et al., 2020; Yuan & Fang, 2020; Shi et al., 2020). Particularly, recent advances in view-based approaches lead to impressive state-of-the-art performances in various 3D vision tasks such as 3D shape classification, retrieval, correspondence, and segmentation (Wei et al., 2020). The most popular techniques often leverage the ground truth labels to train deep neural networks and learn to extract deep representations by aggregating information from a static and pre-selected set of different views. Those approaches, however, often face challenges posed by the requirement of a large amount of annotated

training data and the lack of a mechanism for the adaptive selection of shape-instance-dependent views towards the learning of more informative 3D shape representation.

Figure 1: **Illustration of our pipeline.**

This paper addresses those two challenging issues by introducing the concept of adversarial views (Xiang et al., 2019) and developing a new mechanism to generate views for the adversarial training of a self-supervised model for 3D representation learning, denoted as CoLAV. Specifically, different from the recent advances in multi-view approaches (Kanezaki et al., 2018), CoLAV gains advantages by learning the 3D shape representation without the need for labeled data. In addition, in contrast to using a static set of views as input during the training of a multiview-based 3D shape representation learning framework, we take a further step to investigate the effect of using dynamically rendered views as input during the 3D shape representation learning. To this end, we introduce an adversarial view generator that enables the dynamic generation of shape-instance-dependent adversarial views, which can fool the deep learning model to misidentify its instance-wise identity, as positive pairs to adversarially train robust contrastive learning models towards the learning of more informative 3D shape representation. Our pipeline can be found in Figure 1. Specifically, our model takes a 3D shape instance $x$ as input, and we first generate an augmented 3D shape $t(x)$ by adopting a set of augmentation operations $t$ sampled from a pool of predefined augmentations. Then, we use a randomly initialized a rotation matrix (characterized by $\alpha$, $\beta$, and $\gamma$) to generate rotated 3D shape $x_0^{adv}$. For each 3D instance $x$, $t(x)$, and $x_0^{adv}$ we render $k$ views using the same differentiable renderer $r_\tau(\cdot)$. The rendered views are fed to a descriptor learning module to extract shape descriptors. After we obtained the descriptors, we use a contrastive loss to measure the mutual information of descriptors. Two key components are employed in our 3D shape unsupervised descriptors learning process. The first one is **Adversarial view**. We use the contrastive loss to calculate the gradient with respect to $\alpha$, $\beta$, and $\gamma$ and update the rotation matrix using the updating rule which introduce in the method section. This rotation matrix updating is represented by the purple path. The second component is the **contrastive learning module with a shape-instance-dependent adversarial view attack**, represented in the red path, which is adopted towards the learning of more informative and robust 3D shape representations. In this component, a new 3D instance $x_i^{adv}$, denoted as adversarial 3D instance, can be obtained using the updated rotation matrix. We adopt the same differentiable renderer $r_\tau(\cdot)$ to render adversarial views for the adversarial training of the Descriptor learning module in order to produce robust and informative 3D shape descriptor.

We summarized our contribution as follows: 1) We introduce a novel concept of adversarial views of a 3D shape and develop a new mechanism to adaptively and dynamically render shape-instance-dependent views towards the learning of more informative 3D shape representation via CoLAV. 2)

The proposed CoLAV is capable of learning 3D shape representations from multiple 2D views without requiring ground truth annotations. 3) Our comprehensive experimental results on 3D shape classification and retrieval demonstrate that the 3D shape descriptor learned by CoLAV exhibits superior performance for 3D shape recognition over other techniques, even though CoLAV is completely self-trained with unlabeled 3D datasets (e.g., ModelNet40).

## 2 RELATED WORKS

### 2.1 MULTI-VIEW LEARNING

In recent research, multi-view-based methods have become a popular topic in 3D computer vision. Basically, these methods start with rendering projections of 3D objects from different view perspectives, generate representation features for each projection, and merge them into a global descriptor of the 3D objects. The compact descriptor can be used to improve performance and robustness in 3D object learning tasks such as classification and retrieval. One of the earliest methods in multi-view-based learning is called multi-view convolutional neural network (MVCNN) (Su et al., 2015), which fuses feature representations extracted from multiple views into one compact descriptor by a max-pooling layer. Ruizhongtai Qi et al. (2016) advances the progress by incorporating a multi-resolution component into MVCNN to boost the classification accuracy. However, using a max-pooling layer to fuse representation features from multiple views can only retain the maximum value on each dimension leading to possible losses of information. The group-view convolutional neural network (GVCNN) (Feng et al., 2018) is proposed to solve this problem, which generates a descriptor via a grouping strategy that keeps more information from different views. There is also research attaching attention-based models to the framework, which leverages spatial relationship among different views' feature representation (Han et al., 2018; 2019). Yang & Wang (2019) also leverages the relationship between views and constructs a model that connects corresponding regions on each projection and strengths the representation learned from each individual viewpoint. The aforementioned view-based methods have achieved promising performance on various tasks. However, their training process requires a large amount of annotated data. To alleviate the dependency of annotated data, Ho et al. (2020) proposed a self-supervised method that randomly selects a view feature out of several pre-selected static views of a 3D object as a prototype to learn an invariant stochastic object embedding which is further used to generalize the method (Ho et al., 2020) on test data. Different from all previous view-based methods, we investigate the effect of using dynamic views. Instead of using pre-selected static views, in our work, dynamically rendered views are used to learn an informative multi-view-based 3D object representation, meanwhile, no annotated class labels are given. This encourages the proposed model to better generalize on various unseen views.

### 2.2 ADVERSARIAL LEARNING

In the concept of adversarial learning, adversarial examples are treated as hard positive samples in the model training process to improve the robustness of neural networks (Szegedy et al., 2013). Goodfellow et al. (2014) introduced the fast gradient sign method (FGSM) which increases the robustness of the model by perturbing input samples along with gradient directions. Following this direction, numerous methods (Kurakin et al., 2016; Papernot et al., 2016; Shafahi et al., 2019; Xie & Yuille, 2019; Xie et al., 2020) were proposed and have obtained impressive robustness against adversarial attack in image space. Recently, Zeng et al. (2019) claimed that instead of generating adversarial samples in 2D space, finding adversarial 3D objects behind the projected 2D images is crucial to improve the robustness. In their work (Zeng et al., 2019), FGSM is used to perturb the input sample in 3D space governed by a supervised loss, and ground truth class labels are required to generate adversarial 3D objects. Unlike Zeng et al. (2019), in this work, we introduce a new approach to generate adversarial views based on 3D objects without using ground truth labels.

### 2.3 CONTRASTIVE LEARNING

The concept of contrastive learning has been extensively used in recent studies, especially in self-supervised (SSL) learning scenarios. In SSL 3D object learning, contrastive learning can help retrieve an invariant descriptor for each 3D object by applying random augmentations to each object and minimizing the contrastive loss by measuring the similarity between the feature vectors repre-

senting original and augmented shapes. There have been a certain amount of augmentation strategies that came up by previous studies in the 2D computer vision field (Wu et al., 2018; Ye et al., 2019; Tian et al., 2019), implementing data augmentation by rotation, cropping, and other transformation methods. Such augmentations can force the network to learn sophisticated descriptors to represent input instances while training, thus boosting robustness. By comparing different methods of augmentation, Chen et al. (2020) indicates that the effectiveness of contrastive learning can be enhanced by employing a stronger data augmentation. On the other hand, although we have various augmentation methods to boost performance, it is arduous for contrastive learning models to overcome the hard negative mining problem mentioned in Sohn (2016); Wu et al. (2017); Schroff et al. (2015). One of the possible solutions to that problem is to select negative samples from a distribution of noise that treats all negative instances equally, which is proposed by the contrastive noise estimation (NCE) (Gutmann & Hyvärinen, 2010), by which contrastive learning is directly inspired Bose et al. (2018); Bose et al.; Ho & Vasconcelos (2020).

## 3 METHODS

The pipeline of our proposed module is illustrated in Fig. 1. The CoLAV is a generic module that generates instance-wise adversarial views to adversarially train a contrastive learning model towards the goal of learning a more informative and robust shape descriptor for a given task.

### 3.1 ADVERSARIAL VIEWS

We introduce our novel approach to obtain adversarial views while alleviating the dependency of annotated data. As we illustrated in Section 2, although many multi-view-based methods have been proposed, their performances rely on contents of input views which are controlled to camera positions and annotated data. As we observed from existing works (Su et al., 2015; Feng et al., 2018), camera positions are static across the entire 3D shape dataset, which leads the trained model vulnerable to unseen views. To address this issue, we introduce the concept of adversarial views as hard positive training samples during the model training, which aims to fool the predictor to make wrong predictions by maximizing the objective function.

**Adversarial Views Generation.** Concretely, the generation of dynamic adversarial views depends on the generation of an adversarial 3D sample $x^{adv}$. To generate adversarial 3D samples, existing researches focus on perturbing 3D point positions within a certain range in 3D space. This perturbation is almost indistinguishable by human eyes, i.e., a perturbed 3D sample is visually identical to the original 3D object. Different from the existing adversarial approaches, we propose a method to generate visually different adversarial views of each 3D instance with the aim to fake the model to treat the generated views as new instances. Given a 3D shape $x$ from the dataset $\mathbb{D}$, a differentiable renderer $r_\tau(\cdot) : x \rightarrow I$, a rotation matrix $M$ controlled by three randomly initialized parameters $\alpha$, $\beta$, $\gamma$, a descriptor learning model $f_\theta(\cdot)$ and a task-specific predictor $g_\phi(\cdot)$, the generation of $x^{adv}$ is defined as:

$$x^{adv} = \begin{cases} Mx, & i = 0 \\ \hat{M}x_{i-1}^{adv}, & i > 0, \end{cases}$$ (1)

where $i$ denotes the generation iterations and $\hat{M}$ denotes the updated rotation matrix controlled by $\hat{\alpha}, \hat{\beta}, \hat{\gamma}$ which are formulated as follow:

$$\hat{\alpha}, \hat{\beta}, \hat{\gamma} = p(\epsilon sign(\nabla_{\alpha,\beta,\gamma}\mathcal{L}_{\phi,\theta}(x_{i-1}^{adv}) + [\alpha, \beta, \gamma]),$$ (2)

where $\mathcal{L}$ is the loss for training the neural network $f_\theta(\cdot)$ and $g_\phi(\cdot)$ with parameter $\phi$ and $\theta$, the $sign(\cdot)$ gives the sign of gradients, and $\epsilon$ represents the step size of one gradient attack. We employ a function $p(\cdot)$ as projection function to clip three angles $\hat{\alpha}, \hat{\beta}, \hat{\gamma}$ with in $[-180°, 180°]$. The adversarial views $I^{adv}$ is rendered by:

$$I^{adv} = r(x^{adv})$$ (3)

To train a neural network using adversarial samples generated from Equation 12, we can directly minimize the $\mathcal{L}$ over $I^{adv}$. This adversarial training process can be formulated as follow:

$$\arg\min_{\theta,\phi} \mathbb{E}_{(I) \sim r(\mathbb{D})}[\max_{I^{adv}} \mathcal{L}_{\phi,\theta}]$$ (4)

The computation of loss term $\mathcal{L}_{\phi,\theta}$, such as cross-entropy loss requires the ground truth labels, which prevents us from applying this approach to unannotated datasets. To address this, we alleviate the dependency of ground truth annotations by adopting a self-supervised contrastive loss.

**Self-Supervised Adversarial Views Generation.** Specifically, given a pool of augmentation operations $T$, such as random rotation, random jittering, random scaling, random noise, and random dropout, we randomly select a set of augmentation operations $t \sim T$. The specific data augmentation operation that we adopt in our method can be found in Section 4.2. Our goal is to maximize the similarity between $f_\theta(r(x))$ and $f_\theta(t(r(x)))$, since $r(x)$ and $t(r(x))$ have the same instance-level proprieties, and to minimize the similarity between $f_\theta(t(r(x)))$ and $f_\theta(t(r(x')))$, where $x'$ is a different 3D shape instance. For a clear illustration, we denote $r(x)$ as $I$, $t(r(x))$ as $I_{pos}$, and $t(r(x'))$ as $I_{neg}$. Then, the constrastive learning process can be formulated by minimizing the following loss function:

$$\mathcal{L}_{\mathbf{co},\theta}(I, I_{pos}, I_{neg}) = -\log \frac{e^{(sim(f(I),f(I_{pos}))/\Gamma)}}{e^{sim(f(I),f(I_{pos}))/\Gamma} + e^{sim(f(I),f(I_{neg}))/\Gamma}}, \tag{5}$$

where we follow the SimCLR (Chen et al., 2020) to adopt $\Gamma$ as a temperature parameter and the $sim(\cdot, \cdot)$ function, which is defined as follow:

$$sim(a, b) = a^T b / \|a\| \|b\|. \tag{6}$$

By adopting this formulation, we can reformulate Equation 2 and make it self-supervised, which relaxes the constrain of requiring annotated datasets during adversarial view generation. The self-supervised version of adversarial view generation can be formulated as follow:

$$\hat{\alpha}, \hat{\beta}, \hat{\gamma} = p(\epsilon sign(\nabla_{\alpha,\beta,\gamma} \mathcal{L}_{\mathbf{co},\theta}(r(x), \{r(t(x)), r(t(x_{i-1}^{adv}))\}, r(t(x')))) + [\alpha, \beta, \gamma]), \tag{7}$$

### 3.2 Contrastive Learning with Shape-Instance-Dependent Adversarial View Attack

Once we have the adversarial views as our positive samples, generated from Equation 12, we can learn a robust and informative 3D shape descriptor through contrastive learning. The learning process can now be formulated as follow:

$$\arg\min_\theta \mathbb{E}_{x \sim \mathbb{D}}[\max_{x^{adv}} \mathcal{L}_{\mathbf{3Dco},\theta}(x, \{t(x), t(x^{adv})\}, \{t(x')\})]. \tag{8}$$

The final loss function $\mathcal{L}_{\mathbf{3Dco}}$ can be described as:

$$\mathcal{L}_{\mathbf{3Dco}}(x, \{x_{pos}\}, \{x_{neg}\}) =$$
$$-\log \frac{\sum_{\{x_{pos}\}} e^{(sim(f(r(x)),f(\{r(t(x_{pos}))\}))/\Gamma)}}{\sum_{\{x_{pos}\}} e^{(sim(f(r(x)),f(\{r(t(x_{pos}))\}))/\Gamma)} + \sum_{\{x_{neg}\}} e^{sim(f(r(x)),f(\{r(t(x_{neg}))\}))/\Gamma}}, \tag{9}$$

where $\{x_{pos}\}$ is the positive 3D sample set which is composed by $\{t(x), t(x^{adv})\}$ and $\{x_{neg}\}$ denotes the set of negative 3D samples $\{t(x')\}$. The training of CoLAV can be found in Algorithm 1.

## 4 Experiments

In this section, we conduct experiments to evaluate the performance of our proposed method on two benchmark datasets. More specifically, we evaluate our module when adopting various multi-view-based backbones, including GVCNN (Feng et al., 2018) and MVCNN (Su et al., 2015), as our descriptor learning module. Section 4.1 provides implementation details of our model and a description of datasets used in our experiments. In the following section, we carry out experiments to classify and retrieve 3D objects. In Section 4.3, we demonstrate that the descriptor extracted by our model is robust to rotation. In Section 4.4, we further investigate our model's effectiveness by using different settings, such as different representations of 3D inputs, different numbers of views, etc.

---

**Algorithm 1** The training process of our proposed method

---

**Input:** Original 3D shapes: $x$, augmented 3D shapes: $t(x)$, augmented negative 3D shapes: $t(x')$, descriptor learning model: $f$, parameter of model: $\theta$, adversarial view generation iterations: $i$;
    **for all** iteration $\in$ training stage **do**
        **for all** $x \in$ minibatch **do**
            **for** attack iteration $\in [0, i)$ **do**
                update rotation matrix $M$.
            **end for**
            generate adversarial views.
        **end for**
        calculate objective function $\mathcal{L}_{\textbf{3Dco}}$ using Equation 9.
        update model parameter $\theta$
    **end for**

---

## 4.1 DATASET, IMPLEMENTATION AND TRAINING DETAILS

**Dataset:** We conduct experiments on the ModelNet40 (Wu et al., 2015) dataset. ModelNet40 is one of the first benchmark datasets that were acquired from CAD objects. This dataset contains 12,311 high-resolution, watertight 3D shapes in the mesh format, and all 3D meshes are labeled into 40 categories. In our experiment, we apply the standard splitting of the dataset. We use 9,843 shapes for training and 2,468 shapes for testing. In addition to ModelNet40 benchmark dataset, we also evaluate our method on ScanObjectNN (Uy et al., 2019) dataset. Due to the page limit, we include this part in our supplementary materials.

**Implementation and Training Details:**

The entire model is implemented in Pytorch. An Adam optimizer with a batch size of 16 is used to train the entire network. The momentum was 0.9, and the weight decay rate is set to $10^{-5}$. The learning rate starts at 0.0005 and then decreases by a factor of 0.5 every 20 epochs.

| Method | Modality | Supervision | Classification | Retrieval |
|---|---|:---:|:---:|:---:|
| PointNet Qi et al. (2017a) | point cloud | ✓ | 89.2 | 70.5 |
| PointNet++ Qi et al. (2017b) | point cloud | ✓ | 91.9 | 85.3 |
| PointCNN Li et al. (2018) | point cloud | ✓ | 91.8 | 83.8 |
| Shi. et. al. Shi et al. (2020) | point cloud | ✗ | 84.7 | 71.8 |
| VoxNet Maturana & Scherer (2015) | voxel grid | ✓ | 83.0 | - |
| SubvolumeSup Qi et al. | voxel grid | ✓ | 89.2 | - |
| Voxception Brock et al. (2016) | voxel grid | ✓ | 91.3 | - |
| 3D-DescripNet Xie et al. (2018) | voxel grid | ✗ | 83.8 | - |
| 3D-GAN Wu et al. (2016) | voxel grid | ✗ | 83.3 | - |
| T-L Network Shi et al. (2020) | image + voxel | ✗ | 74.4 | - |
| MVCNN Su et al. (2015) | 12 views | ✓ | 89.9 | 70.1 |
| MVCNN, metric Su et al. (2015) | 12 views | ✓ | 89.5 | 80.2 |
| RotationNet Kanezaki et al. (2018) | 12 views | ✓ | 90.7 | - |
| MHBN Yu et al. (2018) | 12 views | ✓ | 93.4 | - |
| 3D2SeqViews Han et al. (2019) | 12 views | ✓ | 93.4 | **90.8** |
| MVCNN Su et al. (2015) | 12 views | ✓ | 92.2 | 74.1 |
| MVCNN (G.) Su et al. (2015) | 12 views | ✓ | 92.2 | 83.0 |
| GVCNN (G.) Feng et al. (2018) | 12 views | ✓ | 92.6 | 81.3 |
| Ours (MVCNN) | dynamic views | ✗ | **93.6** | 85.3 |
| Ours (GVCNN) | dynamic views | ✗ | **94.3** | 85.9 |

Table 1: **3D Shape classification and shape retrieval results on ModelNet40 dataset.** We compare the classification and retrieval performance of our model with existing methods on the ModelNet40 dataset. Our model is trained without any supervision of ground truth but still outperforms most of the supervised methods on classification accuracy and retrieval mAP. "(Go.)" represents the methods use GoogLeNet as backbone network.

## 4.2 RESULTS ON MODELNET40

**3D Object Classification:**

To the best of our knowledge, our work is the first to extract 3D shape descriptors from dynamic views without using ground truth labels. To validate the generalization ability of our proposed model, we conduct experiments for the task of 3D object classification on the ModelNet40 benchmark and compare our method against various multi-view baed 3D shape descriptor learning methods. In this comparison, we compare not only with multi-view-based methods but also with other types of 3D representation, such as point clouds and voxel grids. For the 3D object classification task,

as part of the inputs, we employ random rotation, random scaling, and random jittering as $t(\cdot)$ to generate $t(x)$. Specifically, for a mesh object $x$, we randomly rotate the object around three axes (x, y, z) and randomly scale the mesh object in the range of $[0.75, 1.15]$. After random rotation and random scaling, we apply random jittering on each vertex of the 3D mesh. For this augmentation operation, we add a random displacement to each point and clip this displacement in the range of $[-0.04.0.04]$. After random jittering, the connection of all faces remains the same as origin 3D instance, which preserves the watertight property of 3D mesh. We use the same data augmentation but apply on another 3D shape instance to produce $t(x')$ as a negative sample. In this experiment, we adopt Pytorch3D as the renderer $r_\tau(\cdot)$ to render 2D views for each 3D instance on-the-fly. We render $k = 12$ images of each 3D shape instance and use $i = 5$ to generate adversarial views. For each attack iterations $i$, we use step size $\epsilon = 0.015$ to update rotation parameters $\alpha, \beta, \gamma$. We use GVCNN and MVCNN as two different backbone networks to generate 3D shape representation $z$. We evaluate the unsupervised learned 3D shape representation $z$ by using a linear SVM with default parameters.

**Experiment Results:** Table 1 shows the classification accuracy of our proposed model as well as other methods on the ModelNet40 dataset. In spite of the fact that our method is completely self-supervised, our MVCNN based model achieves 93.6% accuracy on 3D object classification on the ModelNet40 dataset, which surpasses the classification accuracy of previously proposed supervised methods listed on the table. Our GVCNN based model achieves 94.3% accuracy, which further improves the classification accuracy and surpasses the supervised methods. Compared with MVCNN, Ours(MVCNN) has an improvement of 1.5% on accuracy. Compared with GVCNN, Ours(GVCNN) outperforms GVCNN by 1.7% of accuracy. Another notable difference is that in previous multi-view-based methods on the table, views are all generated statically before training. In our model, new sets of views can be generated for each training epoch. Throughout the entire training process, different views can be dynamically rendered for the training of descriptor learning network to obtain more robust and informative 3D shape descriptors compared with existing methods. To this end, an advantage of dynamically view generation raises, compared with static view-based multi-view methods.

**3D Object Retrieval:** Furthermore, we follow the same training setups that we presented above to conduct the experiment on the task of 3D object retrieval. For each 3D object, its respective shape representation is used to perform the object retrieval. We calculate the distance between every two latent representations to conduct retrieval. Specifically, given a learned 3D shape representation $z$ of a query shape $x$, we aim to find similar shapes by comparing the distance between $z$ and every other shape descriptors. To evaluate the shape retrieval performance, we follow the stander 3D shape retrieval evaluation metric (mAP).

**Experiment Results:** As shown in the last column of Table 1, our MVCNN based self-supervised method obtains 85.3 mAP when performing 3D object retrieval tasks on the ModelNet40 dataset, and GVCNN based method achieves the mAP of 85.9, which exceeds the performance of most of the supervised methods proposed by previous studies. Furthermore, compared to our baseline method MVCNN and GVCNN, our methods improve retrieval mAPs by 2.3 and 4.6 respectively.

## 4.3 ROBUSTNESS TO ROTATION

In this section, we conduct further experiments to explore the resistance to rotation. Since our model dynamically generates 2D views and performs adversarial attacks to force the network to take the worst views into consideration, our model is expected to be highly robust to the rotation. This test

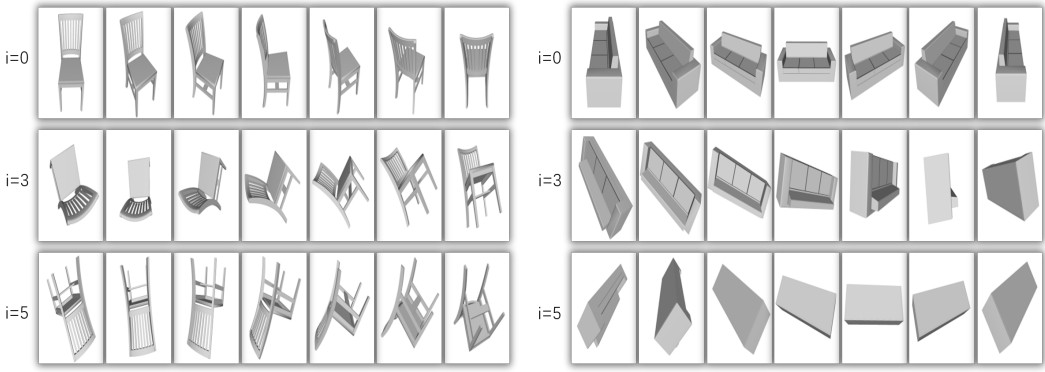

Figure 2: **Adversarial Views:** We observe the adversarial views generated during the adversarial attack procedure. Our model gradually rotates the 3D objects to capture less informative views that can fool the network. These views are utilized to provide difficulty to the training process and urge the network to draw more powerful feature representations.

is designed to measure our performance dealing with 3D objects which are rotated into different angles.

**Experiment Setup:** In this experiment, we test the rotation robustness of our model on the ModelNet40 benchmark dataset. We rotated the testing data around a random axis by either 90 or 180 degrees and reported the performance.

**Experiment Results:** As shown in Table 2 our self-supervised network with both 6 views and 12 views configuration achieves attractive robustness to the rotation. Dealing with 90 degrees of rotation along the gravity axis, our 6 views model performs 90.8 accuracy of classification, which is the best among all the previous methods, including all the supervised methods listed on the table. For 180 degrees of rotation along the Y-axis, our 12 views model outperforms all the previous methods, obtaining 90.9 classification accuracy.

| Method | Modality | Supervision | 90° | 180° |
|---|---|---|---|---|
| PointNet (Qi et al., 2017a) | point cloud | ✓ | 42.5 | 38.6 |
| PointNet++ (Qi et al., 2017b) | point cloud | ✓ | 47.9 | 39.7 |
| RSCNN (Hu et al., 2021) | point cloud | ✓ | 90.3 | 90.3 |
| MVCNN (Su et al., 2015) | 12 views | ✓ | 84.4 | 84.2 |
| ViewGCN (Wei et al., 2020) | 12 views | ✓ | 86.0 | 86.0 |
| GVCNN (Feng et al., 2018) | 12 views | ✓ | 88.9 | 89.7 |
| Ours(MVCNN) | 6 views | ✗ | 89.3 | 89.4 |
| Ours(MVCNN) | 12 views | ✗ | 90.1 | 89.7 |
| Ours(GVCNN) | 6 views | ✗ | **90.8** | 90.3 |
| Ours(GVCNN) | 12 views | ✗ | 90.4 | **90.9** |

Table 2: **Rotation Robustness of 3D Methods.** We test our model's robustness against rotations around Y-axis (vertical to the ground) on ModelNet40. Without using the ground truth supervision, our model with 6 and 12 views configuration performs the highest level of robustness to 90° and 180° rotation among all methods.

## 4.4 ABLATION STUDY

In this section, We study the effect of the components of our model. First, we investigate the effect of the number of views during training and testing, then analysis of the adversarial views generated by our network is presented. Due to the page limitation, the in-depth experiment conducted to explore the effect of different augmentation operations we adopted during the model training process,

the effectiveness of our proposed unsupervised loss and the effectiveness of adversarial view are included in the Appendix.

| Training V. | Testing V. | GVCNN | MVCNN | Ours(GVCNN) | Ours(MVCNN) |
|---|---|---|---|---|---|
| 8 | 2 | 71.2 | 68.3 | 91.2 | 88.7 |
| 8 | 4 | 91.1 | 84.5 | 92.9 | 90.1 |
| 8 | 8 | 93.1 | 87.4 | 93.6 | 91.9 |
| 8 | 12 | 91.5 | 86.8 | 94.1 | 93.1 |
| 12 | 2 | 76.8 | 71.1 | 91.7 | 90.2 |
| 12 | 4 | 90.3 | 87.6 | 92.1 | 91.6 |
| 12 | 8 | 92.1 | 90.5 | 93.9 | 92.2 |
| 12 | 12 | 92.6 | 92.2 | 94.3 | 93.6 |

Table 3: **Effect of Number of Views:** We examine the effect of different numbers of training and testing views. "Training V." indicates the number of views used during the network training, and "Testing V." indicates the number of views used during the inference period. Our network achieves an impressive accuracy even if only two testing views are applied, where "Our (GVCNN)" indicates the GCVNN based network and "Our (MVCNN)" indicates the MVCNN based network.

### 4.4.1 EFFECT OF NUMBER OF VIEWS:

In this study, we investigate the impact of different numbers of views on our model performing 3D object classification. We apply $\{8, 12\}$ views to train our model and retrain the MVCNN and GVCNN. We adopt $\{2, 4, 8, 12\}$ views during the testing phase. As it is listed in Table 3, our model outperforms our baseline GVCNN and MVCNN in all the combinations of training and testing views selected. Even when we employit two views for testing, which is arduous for other methods, our model can still achieve a high level of accuracy. We attribute this success to the adversarial attack procedure that generates challenging positive samples for the network to learn from and causes the network to draw informative descriptors from the two views selected via contrastive learning.

### 4.4.2 ADVERSARIAL VIEW ANALYSIS:

We analyze the effectiveness of our adversarial attack procedure by observing the adversarial views generated by our model. As shown in Figure 2, with $i$ defined in Equation 12 representing the number of iterations of adversarial attack, we can see that our model will gradually rotate the adversarial views to challenging angles which depict less informative details of the 3D object. When $i = 5$, the adversarial views of the sofa are highly ambiguous that even human eyes cannot recognize the object easily. Moreover, with the help of the contrastive learning procedure in the following steps, the network is expected to draw a more sophisticated and invariant descriptor for each 3D object to enhance classification accuracy on these challenging views.

## 5 CONCLUSION

In this paper, we introduce a novel concept of adversarial views generated via a self-supervised contrastive learning mechanism that augments the positive training samples by performing shape-instance-dependent adversarial attacks and introduce a multi-view pipeline that directly takes 3D instance as input. More importantly, different from the previous multi-view-based methods using pre-rendered static views for the training of deep neural networks, our proposed method can dynamically generate views for a given 3D instance on the fly without requiring additional network parameters. Extensive experiments conducted on different benchmark datasets demonstrate the effectiveness of the newly introduced adversarial views. Our self-supervised method boosts the 3D classification accuracy and 3D retrieval mAP to outperform the supervised methods previously proposed. In addition, ablation studies of our model prove the effectiveness and efficiency of the proposed components. Our method highly improves robustness to rotation thanks to the adversarial views dynamically generated by our adversarial attack procedure.

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

# A APPENDIX

## A.1 DETAILED IMPLEMENTATION

We use the Adam optimizer with a batch size of 16 is used to train the entire network. The momentum was 0.9, and the weight decay rate is set to $10^{-5}$. The learning rate starts at 0.0005 and then decreases by a factor of 0.5 every 20 epochs. We train the entire network in 180 epochs and we use a 1024-d latent vector to represent the learned descriptor. To evaluate the shape retrieval scores, we extract 3D descriptors using the trained network. We sort the most relevant shapes for each query by cosine distance and compute the mAP score.

## A.2 3D SHAPE RETRIEVAL TASK ON MODELNET40

We show examples of our 3D object retrieval in Figure 3. For each group of 3D shapes, the shape to the left of the vertical line is the query object from the test set, and on the right side of the line, we show the retrieved objects from the training set.

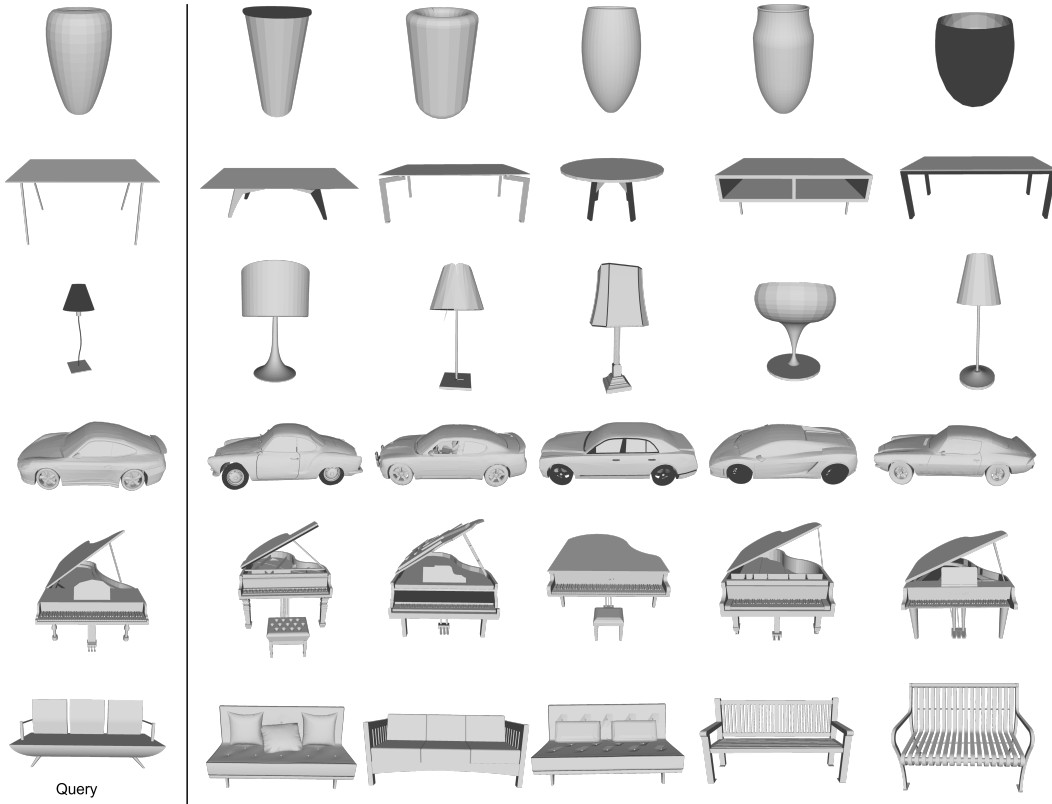

Query

Figure 3: **Qualitative results of 3D shape retrieval task on ModelNet40 dataset:**

## A.3 SCANOBJECTNN.

In addition to ModelNet40 benchmark dataset, we also evaluate our method on ScanObjectNN Uy et al. (2019) dataset. ScanObjectNN is the newest and one of the most challenging datasets for 3D object recognition. It is acquired from the real-world objects in 3D point cloud representation and three variants are proposed for different levels of difficulty. 1) Vanilla ScanObjectNN dataset contains clean 3D object point clouds. 2) Background ScanObjectNN dataset which includes real-world background information of each 3D object. 3) Perturbed ScanObjectNN dataset performs random

| Method | Modality | Supervision | Classification | Retrieval |
|---|---|---|---|---|
| PointNet | point cloud | ✓ | 79.2 | 64.8 |
| PointNet++ | point cloud | ✓ | 84.3 | 76.6 |
| KD-Networks | point cloud | ✓ | 84.1 | 76.1 |
| Shi. et. al. | point cloud | ✗ | 78.7 | 65.1 |
| MVCNN | 12 views | ✓ | 79.8 | 63.9 |
| MVCNN, metric | 12 views | ✓ | 80.1 | 64.1 |
| RotationNet | 12 views | ✓ | 81.4 | - |
| MHBN | 12 views | ✓ | 83.6 | - |
| 3D2SeqViews | 12 views | ✓ | 83.8 | 75.5 |
| MVCNN (GoogLeNet) | 12 views | ✓ | 82.0 | 74.1 |
| MVCNN (GoogLeNet), metric | 12 views | ✓ | 82.1 | 74.3 |
| GVCNN (GoogLeNet) | 12 views | ✓ | 83.1 | 75.9 |
| Ours (MVCNN) | dynamic views | ✗ | **84.1** | **76.2** |
| Ours (GVCNN) | dynamic views | ✗ | **85.4** | **76.8** |

Table 4: **3D Shape classification and retrieval results on ScanObjectNN dataset.** We compare the classification and retrieval performance of our model with previous methods on the ScanObjectNN dataset. Our model is trained without any supervision of ground truth but still outperforms most of the supervised methods on classification accuracy.

perturbation on the Background ScanObjectNN dataset to increase the difficulty of 3D object recognition tasks. In our experiment, we adopt the Vanilla ScanObjectNN dataset to evaluate our method. The reason why we use the "vanilla" version is that other versions of ScanObjectNN contain backgrounds which could lead the propped adversarial view only to reflect the background and occlude the 3D objects. The first variant is composed of 2902 3D point clouds and each 3D point cloud has its unique label from 1 to 15. ScanObjectNN (Uy et al., 2019) randomly split into a training set (80%) and test set (20%) and we follow this protocol to conduct our experiment.

**3D Object Classification:** In this section, we conduct experiments for the task of 3D object classification on the ScanObjectNN benchmark and compare our method with many other methods retrained on the ScanObjectNN dataset. In this comparison, we consider multiple representation learning methods taking different 3D modalities as input, such as point clouds, into consideration. Similar to Section 4.2, we use GVCNN and MVCNN as two different backbone modules $f_\theta(\cdot)$. Different from ModelNet40, ScanObjectNN is a point-cloud-based dataset. To apply the multi-view-based 3D representation learning method, we re-render 3D objects from multiple viewpoints. Here, we adopt 12 viewpoints in this experiment. An elevation angle of $\pi/9$ is applied for all viewpoints with azimuths sampled form $[0, \pi)$ uniformly.

In our proposed model, we adopt Pytorch3D as a differentiable renderer and use its point cloud rendering utilities. We also employ random rotation, random scaling, and random jittering operations as $t/(\cdot)$ to generate $t(x)$ as we specified in Section 4.2. Furthermore, we add the random dropout in $t(\cdot)$ operation as the last augmentation. In this experiment, we use $i = 8$ to perform the adversarial attack with step size $\epsilon = 0.005$ to update rotation parameters $\alpha, \beta, \gamma$. We adopt the same default linear SVM from the scikit-learn library to perform the downstream task of 3D object classification.

**Experiment Results:** In Table 4, we compare the classification accuracy of our model with other retrained methods, including PointNet (Qi et al., 2017a), PointNet++ (Qi et al., 2017b), KD-Networks (Klokov & Lempitsky, 2017), VoxNet (Maturana & Scherer, 2015), 3D-GAN (Wu et al., 2016), MVCNN (Su et al., 2015) and GVCNN (Feng et al., 2018), on the ScanObjectNN dataset. As listed in Table 4, Our two backbones achieve $84.1\%$ and $85.4\%$ of accuracy on 3D object classification, which outperforms existing supervised methods. This demonstrates our proposed method can effectively find challenging views and hence increase the generalize ability of the backbone model. Specifically, compared with MVCNN, our Ours(MVCNN) improves the performance by 2% on accuracy and our Ours(GCVNN) network outperforms GVCNN by 2.3% of accuracy.

**3D Object Retrieval** In this subsection, we follow the protocol that was presented in Section 4.2 to evaluate our proposed method on 3D object retrieval.

**Experiment Results:** As shown in the last column of Table 4, our methods achieve the best results compared with other methods listed in the table. Specifically, Ours(MVCNN) improves the mAP by 1.9 compared with MVCNN (GoogLeNet), metric (Su et al., 2015). Ours(GVCNN) achieves the retrieval mAP of 76.8 which outperform the baseline result, GVCNN (GoogLeNet) (Feng et al., 2018) 75.9, by 0.9. This improvement validates the effectiveness of our proposed method in the 3D object recognition task.

A.4   EFFECT OF DIFFERENT AUGMENTATION OPERATION:

In section 3.3, we introduce our unsupervised method which leverages the power of contrastive learning which relies on different data augmentation operations. In this section, we conduct various experiments on ModelNet40 to investigate the effect of different data augmentations. As presented in Table 5, we adopt two different sets of data augmentation operations for mesh objects. The first set (Set 1) of operations contains random rotation, random scaling, and random jittering. The second set (Set 2) of operations contains random scaling and random jittering. The third set (Set 3) of operations only contains random jittering. In addition to mesh objects, we also conduct experiments on point clouds, and three sets of augmentations are used. Moreover, a random dropout of point cloud has also been added to $t(\cdot)$. To validate the performance on the point cloud, we first randomly sample 2048 points from the ModelNet40 data set, and we use the point cloud rendering function from the Pytorch3D library to render 3D point clouds. From Table 5 we can see that our proposed method achieves the best result when adopting complex data augmentations. Observing from the second and fourth rows and eighth and tenth rows of Table 5 we find that the rotation operation contributes less during the learning process. This can be attributed to our proposed adversarial attack which can generate rotated adversarial samples.

| Method | Modality | Augmentations | Accuracy |
|---|---|---|---|
| Ours(MVCNN) | Mesh | Set 1 | 93.6 |
| Ours(GVCNN) | Mesh | Set 1 | **94.3** |
| Ours(MVCNN) | Mesh | Set 2 | 93.5 |
| Ours(GVCNN) | Mesh | Set 2 | **94.3** |
| Ours(MVCNN) | Mesh | Set 3 | 93.1 |
| Ours(GVCNN) | Mesh | Set 3 | 94.1 |
| Ours(MVCNN) | Point Cloud | Set 1+dropout | 93.1 |
| Ours(GVCNN) | Point Cloud | Set 1+dropout | **93.8** |
| Ours(MVCNN) | Point Cloud | Set 2+dropout | 93.1 |
| Ours(GVCNN) | Point Cloud | Set 2+dropout | **93.78** |
| Ours(MVCNN | Point Cloud | Set 3+dropout | 92.6 |
| Ours(GVCNN) | Point Cloud | Set 3+dropout | 93.2 |

Table 5: **Effect of Different Augmentation** We investigate the effect of different augmentations that are adopted during our experiment. Our model is insensitive to rotation augmentation.

| Method | Modality | Accuracy |
|---|---|---|
| Ours(MVCNN) | Mesh | 93.6 |
| Self-Supervised MV. | Mesh | 92.3 |
| Ours(GVCNN) | Mesh | **94.3** |
| Self-Supervised GV. | Mesh | 93.2 |
| Ours(MVCNN) | Point Cloud | 93.1 |
| Self-Supervised MV. | Point Cloud | 91.8 |
| Ours(GVCNN) | Point Cloud | **93.8** |
| Self-Supervised GV. | Point Cloud | 92.7 |

Table 6: **Effect of Unsupervised Loss** We investigate the effect of our proposed adversarial attack by comparing it with a supervised version.

A.4.1 EFFECT OF UNSUPERVISED LOSS:

In this section, we further investigate the effectiveness of our proposed adversarial attack from two aspects. We train Ours(GVCNN) on the ModelNet40 dataset using 9,843 shapes and compare our proposed method with our method that depreciates the adversarial attack which is a self-supervised multi-view-based 3D representation learning method. Specifically, we use random rotation, random scaling, and random jittering as $t(\cdot)$, and we use

$$\mathcal{L}_{\textbf{3dco},\theta}(x, t(x), t(x')) = \tag{10}$$
$$-\log \frac{e^{(sim(f(r(x)),f(r(t(x))))/\Gamma}}{e^{sim(f(r(x)),f(r(t(x))))/\Gamma} + e^{sim(f(r(x)),f(r(t(x'))))/\Gamma}},$$

as the objective function to unsupervisedly train the network. The experimental results can be found in Table 6. Our adversarial attack achieved over 1.0% of improvement on classification accuracy comparing with the self-supervised approach. Moreover, our supervised methods also outperform GVCNN and MVCNN by 0.4% and 0.1% respectively.

A.5 EFFECT OF ADVERSARIAL VIEWS:

we further conduct an experiment by applying random rotation (as opposed to the gradient-descent guided rotation) to input 3D instance to generate adversarial views to verify the effectiveness of our proposed adversarial views. Specifically, we modify Equation 7 as:

$$\hat{\alpha}, \hat{\beta}, \hat{\gamma} = (p(\Delta\alpha, \Delta\beta, \Delta\gamma) + [\alpha, \beta, \gamma]), \tag{11}$$

and we follow

$$x^{adv} = \begin{cases} Mx, & i = 0 \\ \hat{M}x_{i-1}^{adv}, & i > 0, \end{cases} \tag{12}$$

to generate views by using random $\Delta\alpha, \Delta\beta, \Delta\gamma$ for each step of adversarial attack, where $M$ is a random initialized rotation matrix, and $\hat{M}$ is a rotation matrix controlled by $\hat{\alpha}, \hat{\beta}, \hat{\gamma}$. We use GVCNN as the backbone network and follow the same experiment setting in our main paper to train the new module. We report the classification accuracy on ModelNet40, denoted as "Random rotation (multiple steps)". Moreover, we compare our model with the one without using adversarial views. Specifically, we replace the adversarial view generation from our method with a random rotation operation, denoted as "No adversarial views generation", to validate its effectiveness. As

| Method | Classification acc. |
|---|---|
| No adversarial views generation | 91.4 |
| Random rotation (multiple steps) | 92.3 |
| Ours | **94.3** |

Table 7: **Effect of Adversarial Views.** Comparison between our method and random rotation in adversarial view generation.

presented in this table, our method outperforms other two methods by at least 2%, which proves the effectiveness of our proposed adversarial view generation method.

In addition to the aforementioned experiment, we conduct an experiment on the number of back-propagation steps for the rotation matrix to answer this question. We use $i = 1, 3, 5, 7$ while the other experimental settings remain the same as we presented in Section 4 of the main submission to validate the effectiveness of our method. As listed in Table 8, our method reaches the best performance when we set i=5 during the training process.

A.6 COMPUTATIONAL OVERHEAD:

We conducted an additional experiment as follows. We first generated a collection of 2D view images offline through rending the 3D objects sampled from ModelNet40. Then we compared the computational overhead for the training with the offline rendered view images and for the training with online view generation (our method). We report the time of a forward pass for one input

| Method | Classification Acc. |
|--------|---------------------|
| Ours-1 | 92.4 |
| Ours-3 | 93.8 |
| Ours-5 | **94.3** |
| Ours-7 | 94.2 |

Table 8: **Effect of Adversarial Attack Steps.** Ours-$i$ represents the number of adversarial attack adopted during training.

| Method | Time |
|--------|------|
| MVCNN | 39.89ms |
| GVCNN | 40.12ms |
| Ours(MVCNN) | 43.39ms |
| Ours(GVCNN) | 43.62ms |

Table 9: **Computational Overhead Comparison.** Please note the time reported in this table for MVCNN and GVCNN exclude the time cost by the offline rendering process. Ours(MVCNN) and Ours(GVCNN) denotes the different backbone networks.

sample. Due to the differentiable rendering process is implemented under the Pyotroch framework, the rendering process can be accelerated by GPU which can significantly increase the rendering speed. As we can see from the experimental results, our online adversarial views generation during the training does not introduce a significant computational overhead.

