# OpenReview forum: "Contrastive Learning of 3D Shape Descriptor with Dynamic Adversarial Views"
_ICLR.cc/2022/Conference — ICLR 2022 Submitted_

### Official Review · Reviewer_aAcK · 2021-10-31

**Correctness:** 3
**Technical Novelty And Significance:** 2
**Empirical Novelty And Significance:** 2
**Recommendation:** 5
**Confidence:** 4

**Main Review:**

Strengths:
- Exploiting a differentiable renderer to optimize for the viewpoints that fool the network the most seems to be effective, as displayed on Fig 2.
- The method is applied to 2 different backbone networks (MVCNN and GVCNN) and is thus generic.
- Metrics are indeed in favour of the proposed method compared to a supervised training, with no augmentation or adversarial samples.


Weaknesses:
- The idea of using gradient descent for optimising for the pose of objects that fool a CNN is not novel, and references should be added. For example, see *Strike (with) a Pose: Neural Networks Are Easily Fooled by Strange Poses of Familiar Objects* (Alcorn et al., CVPR 2019).
- The benefit of 3D augmentation and adversarial view sampling should clearly be investigated separately. The very last table of the supplementary (Tab. 6) is an attempt at ablating the adversarial view generation but should be made clearer (why is the caption mentioning a “supervised version”?) and moved to the main paper. The impact on final performance on the number of back propagation steps for the rotation matrix would also contribute to prove the method is effective. Similarly, Table 5 in the supplementary is an attempt at quantifying the effect of augmentations, but seems to be carried on already trained networks. What would be interesting is to train networks with different sets of augmentation and describe which augmentations are the most effective.
- Figure 1 is unclear, and might benefit from being split into 2. Currently it is trying to describe both contrastive learning and adversarial view generation, but both are mixed and therefore not clearly displayed. For example:
    - Why is a differentiable renderer needed for the augmented instance and the input 3D instance? Having a differentiable renderer is only useful for the adversarial instance.
    - The caption for “adversarial view generation” introduces the augmented 3D shape $t(x)$, while the adversarial view is not related to this augmentation.
    - Similarly, a “rotation matrix” is mentioned, but not introduced previously
- By contrast to the above points, some experiments are displayed in the main paper that are of less relevance, such as “4.4.1 effect of the number of views”: this does not help to quantify the benefits of the proposed method.
- In Fig. 2, it would be interesting to show how L is increasing during hard sample mining.

- The conclusion mentions a “novel multi-view pipeline”, but the pipeline/network is not novel. What is novel is a way of augmenting data during training, and using a contrastive loss on it.

Small missing technical details:
- What is the number of training epochs?
- What is the size of a descriptor?
- How is the task of object retrieval carried on?

Many typos, and quite verbose language that does not help having a smooth read:
- The abstract+intro should state more clearly what the contributions are. The “selection of shape-instance-dependent views towards the learning of more informative 3D shape representation” is not really informative.
- Fig. 1 “prospective”?
- 2.2 “required to generatING adversarial…”
- 3.1 “To this end, the direct adaptation of the adversarial sample generation method prevents us dynamically generating adversarial views due to field viewpoints” is unclear.
- In the very next sentence the differentiable renderer is introduced twice, and a task specify predictor is introduced but was never mentioned before.
- Equation (2) refers to t and x’ which were not introduced previously.
- “Self-supervised Adversarial Views Generation” refers to Sections ?? and ??, and mentions “proprieties” instead of “property”. “We can reformulate Equation ???”
- 3.2 : “generated from Equation 1, 3, 7” : pick one? Similarly in Algorithm 1: pick one equation and make it self contained?
- Algorithm 1 refers to L_final, which was not introduced previously.
- 4.1 Dataset “datasets that WERE acquired” / “WATERTIGHT 3D shapes”
- 4.1 Implementation and training details: “We adopt a pytorch 3d” -> “We use pytorch3d” . “One-the-fly” -> on-the-fly
- 4.2 “Random mash” -> mesh. “Connoations of all feces” -> sorry what? “Propriety” -> property.
- End of Experiment results “ To this end, a significant advantage of our proposed method raises compared” : unclear.
- …
- The supplementary has broken references to the main paper (“Section ??”)

**Summary Of The Paper:**

The authors train a 3D object descriptor using existing multi view CNN architectures, but improve the training procedure with:
- augmentations in the form of 3D rotations,
- adversarial view sampling for forging hard examples during training,
- an unsupervised (contrastive) loss.

When trained with this procedure, CNNs are more performant on downstream tasks such as object retrieval or classification.

**Summary Of The Review:**

The method seems to be working well, but is not very novel and lacks ablation studies to disentangle which components contribute to the performance improvements and how.

---

> ### Author Response · Authors · 2021-11-23
> **Response to Reviewer aAcK (Part 1)**
>
> **Q1:** The idea of using gradient descent for optimising for the pose of objects that fool a CNN is not novel, and references should be added. For example, see Strike (with) a Pose: Neural Networks Are Easily Fooled by Strange Poses of Familiar Objects (Alcorn et al., CVPR 2019).
>
> **A1:** Gradient descent based adversarial attack for optimization is a commonly used mechanism in various computer vision researches [1,2] to enhance the robustness of machine learning models.
> In our work, we take advantage of this adversarial learning mechanism in view-based 3D shape descriptor learning.
> It is known that view-based 3D shape descriptor learning methods face challenges posed by the lack of mechanisms for the adaptive selection of expressive views towards the learning of robust and informative 3D shape descriptors. We address this challenging issue by introducing the novel concept of adversarial views.
>
> To the best of our knowledge, it is the first time that gradient descent based adversarial learning is introduced to find 2D adversarial views through the process of mining hard negative samples for training. The mined adversarial views are further used to adversarially train a deep neural network to learn a robust and discriminative 3D descriptor. This proposed adversarial view-based 3D shape descriptors learning process is novel in 3D shape analysis and also proved to be effective in our paper. The comprehensive experiments reported in our revised version prove the effectiveness of our proposed model.
>
>
> **Q2:** Figure 1 is unclear, and might benefit from being split into 2. Currently it is trying to describe both contrastive learning and adversarial view generation, but both are mixed and therefore not clearly displayed. For example: Why is a differentiable renderer needed for the augmented instance and the input 3D instance? Having a differentiable renderer is only useful for the adversarial instance.
> The caption for “adversarial view generation” introduces the augmented 3D shape , while the adversarial view is not related to this augmentation.
> Similarly, a “rotation matrix” is mentioned, but not introduced previously
>
> **A2:** Thank you for your advice regarding Figure 1. We would like to first clarify the reason why we use a differentiable renderer for the augmented 3D instance and the input 3D instance. By using the same differentiable renderer for the input, the augmented, and the adversarial instances, we can ensure the domain consistency among the rendered views from three types of instances. With the same renderer used for all 3D instances, it naturally prevents the domain discrepancy introduced by using different types of renderers. Please note that the adversarial 3D instance is dynamically rotating in reference to the cameras driven by the gradient back-propagated from the adversarial views to the rotation matrix through the differentiable renderer. In contrast, the augmented and the input 3D instances are fixed in reference to the cameras during the training process without back-propagating gradients through the differentiable renderer. In addition to this specific question, we have also updated Figure 1, in the revised version of our submission for a clear presentation. We add a detailed description of our pipeline at the end of the introduction section of the revised paper.
>
> **Q3:** The impact on final performance on the number of back propagation steps for the rotation matrix would also contribute to prove the method is effective
>
> **A3:** We conduct an experiment on the number of back-propagation steps for the rotation matrix to answer this question. We use $i={1,3,5,7}$ while the other experimental settings remain the same as we presented in Section 4 of the main submission to validate the effectiveness of our method. This experiment can also be found in the supplementary material of our revised version.
>
> |Method|Classification Acc.|
> |---|---|
> |Ours-1      |92.4   |
> |Ours-3      |93.8  |
> |Ours-5      |**94.3**   |
> |Ours-7     |94.2   |
>
> As listed in this table, our method reaches the best performance when we set i=5 during the training process.
>
> ```
> [1] Michael A Alcorn, Qi Li, Zhitao Gong, Chengfei Wang, Long Mai, Wei-Shinn Ku, and Anh Nguyen. Strike (with) a pose: Neural networks are easily fooled by strange poses of familiar objects. In Proceedings of the IEEE/CVF Conference on Computer Vision and Pattern Recognition, pp.4845–4854, 2019
>
> [2] Jiawei Su, Danilo Vasconcellos Vargas, and Kouichi Sakurai. One pixel attack for fooling deep neural networks. IEEE Transactions on Evolutionary Computation, 23(5):828–841, 2019
> ```

---

> ### Author Response · Authors · 2021-11-23
> **Response to Reviewer aAcK (Part 2)**
>
> **Q4:** The caption for “adversarial view generation” introduces the augmented 3D shape, while the adversarial view is not related to this augmentation.
>
> **A4:** We apologize for the misleading. We have added an explanation of how the augmented 3D instance related to adversarial views generation in the introduction section of the revised manuscript. We also provide our rationale here. In our framework, the augmented 3D shape is used as positive samples in contrastive learning to form the contrastive loss function. The gradient of the loss function is then back-propagated along the ascent direction as the form of adversarial attack to determine a rotation matrix. The rotation matrix is applied to the input 3D instance to generate an adversarial 3D instance, which is further used to generate adversarial views through the differentiable renderer. Hence, the adversarial view is related to augmented 3D instances.
>
> We also would like to answer this question in detail by presenting several equations listed in Section 3.1 of our paper. The adversarial view is generated by using
>
> $$
> x^{adv}=
> \begin{cases}
> Mx& \text{i=0} \\\\
> \hat{M}x^{adv}_{i-1}& \text{i>0}
> \end{cases}
> $$
>
> and
>
> $$
>     \hat{\alpha},\hat{\beta},\hat{\gamma}= p(\epsilon sign(\nabla_{\alpha,\beta,\gamma} L_{3Dco,\theta}(r(x),\{r(\textbf{t(x)}),r(t(x^{adv}_{i-1}))\},r(t(x')))+[\alpha,\beta,\gamma])
> $$
>
> where the $x$ is input 3D instance, $x^{adv}$ is the adversarial 3D instance,  $M$ is a rotation matrix controlled by $\alpha,\beta,\gamma$, and $\hat{M}$ is the updated rotation matrix controlled by $\hat{\alpha},\hat{\beta},\hat{\gamma}$. As we can see from the last equation, the bolded term is the augmented 3D instance which is related to the generation of $x^{adv}$. The adversarial instance is rendered into 2D adversarial views by using:
>
> $$I^{adv}=r(x^{adv})$$
>
> where $r()$ is the differentiable renderer. Therefore the augmented 3D instance $t(x)$ is related to adversarial views.
>
>
> **Q5:** Small missing technical details: What is the number of training epochs? What is the size of a descriptor? How is the task of object retrieval carried on?
>
> **A5:** We use the Adam optimizer with a batch size of 16 to train the entire network. The momentum is 0.9, and the weight decay rate is set to $10^{-5}$. The learning rate starts at 0.0005 and then decreases by a factor of 0.5 every 20 epochs. We train the entire network for 180 epochs and we use a 1024-d latent vector to represent the learned descriptor. We extract 3D descriptors using the trained network and we follow the stander evaluation metrics (i.e., mAP scores) of 3D shape retrieval to evaluate the performance.
>
> **Q6:** By contrast to the above points, some experiments are displayed in the main paper that are of less relevance.
>
> **A6:**  Thank you for your valuable comments. We respect your comment, we will move the less relevant experiment to the supplementary material of our revised version as recommended. In addition, as suggested by other reviewers, we also added several extra experiments to validate the effectiveness of our proposed methods. Specifically, we first conduct an experiment to verify the effectiveness of our proposed adversarial views by comparing the classification accuracy of our proposed method and other data augmentation methods. In addition, we conduct an experiment to show the computational overhead of our method. Please refer to the revised version for the detailed experiment.
>
> **Q7:** The conclusion mentions a “novel multi-view pipeline”, but the pipeline/network is not novel. What is novel is a way of augmenting data during training, and using a contrastive loss on it.
>
> **A7:** Thank you for your insightful comments, and we agree that the key novelty comes from the novel way of dynamically generating adversarial views as hard negative samples as data augmentation during training. We have revised this in our manuscript accordingly.

---

### Official Review · Reviewer_GNMf · 2021-11-01

**Correctness:** 3
**Technical Novelty And Significance:** 3
**Empirical Novelty And Significance:** 3
**Recommendation:** 6
**Confidence:** 3

**Main Review:**

### Pros
The method is based on two powerful and novel ideas:
- using contrastive learning for learning efficient descriptors in a self-supervised way for this task. The self-supervision alone performs impressively well.
- further improving the method by generating novel adversarial views.

Remarkably, the method outperforms relevant baselines that are trained in a supervised way.

The idea of using adversarial views is particularly interesting: in addition to being efficient, adversarial views are conveniently interpretable contrary to imperceptible noise values in more common adversarial methods. They could be useful for other applications in 3D vision due to their interpretability.

The paper provides comprehensive experiments on multiple datasets while comparing to relevant baselines. Moreover, ablation studies are run extensively on the different parameters and losses.

Up to the results section, the paper is well written and clear. The illustrations are clear and nicely describe the methods and results.

### Cons

The rule for bold numbers in the tables should be consistent. In table 1, the results in bold are the ones from the presented method (“ours”) (3D2SeqViews/Retrieval is not in bold in table 1). In table 2 the best results are in bold.

Starting from section 4.1 the writing feels rushed and is unclear. It would be good to perform a careful pass over this section. Some sentences are unclear and there are many typos.

A list of a few problems from the text:

Section 3.1:
- “Section ?? and Section ??” the section numbers are missing
- “we can reformulate Equation and make” the equation number is missing

Section 4.1:
- water eight 3D shapes in the mesh format -> watertight (?) 3D shapes in the mesh format
- we random rotate -> we randomly rotate
- random scaling the mash object -> randomly scale the mesh
- “Connotations of all feces remain the same as origin”  this sentence is unclear and there is a typo
- water-tight propriety -> watertight property


**Summary Of The Paper:**

The paper proposes a novel method that combines contrastive learning and an adversarial method to improve multi-view  approaches for shape classification and retrieval in a self-supervised way. The authors introduce adversarial views that enable to explicitly train the network on challenging views of a shape. The method is trained in a self-supervised way and does not require any label.

**Summary Of The Review:**

The authors propose powerful and novel ideas. They are clearly supported by the performance shown in the extensive experiments and ablation studies. I strongly suggest the authors to work on the writing, especially in the results section.

---

> ### Author Response · Authors · 2021-11-23
> **Response to Reviewer GNMf**
>
> We appreciate the time and efforts you have spent to deeply understand the key contributions of our paper and provide valuable and positive comments. We have addressed your concerns by answering the following questions
>
> **Q1:** The rule for bold numbers in the tables should be consistent. In table 1, the results in bold are the ones from the presented method (“ours”) (3D2SeqViews/Retrieval is not in bold in table 1). In table 2 the best results are in bold.
>
> **A1:** We apologize for the inconsistency during the presentation of the quantitative results. We have bolded the best performance in Table 1 for a consistent presentation in the revised version.
>
>
> **Q2:** Starting from section 4.1 the writing feels rushed and is unclear. It would be good to perform a careful pass over this section. Some sentences are unclear and there are many typos.
>
> **A2:** To our best, we have proofread the original submission and we carefully revised the paper, and corrected all grammar and writing issues for a better and clearer presentation in the revised version.

---

> > ### Comment · Reviewer_GNMf · 2021-11-29
> > **Thanks**
> >
> > I thank the authors for their response and the efforts in the updated version. My main concerns have been addressed.

---

> > > ### Author Response · Authors · 2021-11-29
> > > **Re: Thanks (ICLR 2022 Conference Paper3813 Reviewer GNMf)**
> > >
> > > Thank you so much for your encouraging comments.

---

### Official Review · Reviewer_6v1k · 2021-11-01

**Correctness:** 2
**Technical Novelty And Significance:** 2
**Empirical Novelty And Significance:** 2
**Recommendation:** 3
**Confidence:** 4

**Main Review:**

The area in which the present paper contributes is important and interesting. The paper proposes an unsupervised descriptor which seems to be efficient.

However, I believe in the current form the paper is not ready for a publication.

**1. The presentation is lacking clarity and details.** For example, in section 3.1. Equation 2 uses $t(\cdot)$ which is not defined until the self-supervised adversarial view generation paragraph. Then $f_{\theta}$ is introduced prior to eq 2 but not used before eq 3, differential renderer is introduced twice before eq 1. There are missing pointers to sections and equations, notation is not strict $Iadv$ vs $I^{adv}$. All these issues undermine the quality of the paper. Then, in the first sentence of section 4, they say they evaluate on 3 dataset, while in fact they do so on two. Table 1 misses references to baselines, what is VoxNet?. The reader can find VoxNet in references and in supplement, but the name VoxNet is not used in the paper apart from the table 1. Furthermore, important implementation details are missing. It's not clear how they integrate into GVCNN and MVCNN, I guess there can be many ways. I believe, this single concern is sufficient to recommend rejection and encourage resubmission.

**2. Unsupervisedness of the approach.** I cannot really name the current approach unsupervised for object classification at least. MVCNN and GVCNN use labels and it's not clear how it's the descriptors are integrated, whether they're finetuned or anything.

**3. Ablation of adversarial views.** It's not totally clear from the paper if the method is better than just using random augmentations and random rotations, or somehow uniformly distributed.

**Summary Of The Paper:**

The paper discusses a method to learning a 3D shape descriptor. They start with meshes of objects, which they render using a differentiable renderer. They pass the rendered images to a feature extraction network $f_{\theta}$ and obtain a feature embedding. This embedding is used in the contrastive learning procedure. Together with the "real" images, they use adversarially perturbed images. The perturbation parameters include rotation angles and random non-linear transformations. At test time they use many images of the object to get object embedding.

The paper is evaluated on one dataset in the main paper and on another one in the supplement. Two problems are considered: object classification and object retrieval. According to the tables the description can bump the accuracy by a couple of percent.

**Summary Of The Review:**

Overall, I believe the paper is below the bar

---

> ### Author Response · Authors · 2021-11-23
> **Response to Reviewer 6v1k**
>
> **Q1:** Unsupervisedness of the approach.
>
> **A1:** Thanks for your valuable comments regarding unsupervised learning of our 3D shape descriptor. We would like to first clarify that the focus of our paper is not unsupervised learning for 3D object classification. The “unsupervisedness of our approach" particularly means that the 3D shape descriptor is learned without any supervision information (i.e. the category labels). Our learned 3D shape descriptor with contrastive learning and adversarial views is NOT fine-tuned during the downstream task of the 3D shape classification process. Similar methods for unsupervised learning of 3D shape descriptors have been proposed in [1,2] which do not require category labels for training. In addition, we would like to iterate that similar to other approaches for unsupervised learned descriptors [1,2], we evaluated our descriptor through the 3D object classification experiments as described in Sections 4 of our original submission. Specifically, similar to [1], we use a linear SVM from the scikit-learn library to measure the performance of our proposed unsupervised shape descriptors.
>
> **Q2:** Ablation of adversarial views.
>
> **A2:** Thank you for the comments. Actually, in the supplementary material of our original submission, we included the related experimental results about the comparison with other data augmentation methods. The comparison results can be found in Table 5 of the supplementary submission. In addition, we further conduct an experiment by applying random rotation (as opposed to the gradient-ascent guided rotation) to input 3D instances to generate random adversarial views to verify the effectiveness of our proposed adversarial views. Specifically, we modify Equation 7 as:
> $$ \hat{\alpha},\hat{\beta},\hat{\gamma}= (p(\Delta\alpha,\Delta\beta,\Delta\gamma)+[\alpha,\beta,\gamma]),$$
> and we follow
> $$
> x^{adv}=
> \begin{cases}
> Mx& \text{i=0} \\\\
> \hat{M}x^{adv}_{i-1}& \text{i>0}
> \end{cases}
> $$
> to generate views by using random $\Delta\alpha,\Delta\beta,\Delta\gamma$ for each step of adversarial attack, where $M$ is a random initialized rotation matrix, and $\hat{M}$ is a rotation matrix controlled by $\hat{\alpha},\hat{\beta},\hat{\gamma}$. We use GVCNN as the backbone network and follow the same experimental settings in our main paper to train the new model. We report the classification accuracy on ModelNet40, denoted as "Random adversarial views''. Moreover, we compare our model with the one without using adversarial views. Specifically, we replace the adversarial view generation from our method with a random rotation operation (a single step), denoted as "No adversarial views generation'', to validate its effectiveness.
>
> |  Method  | Classification Acc.|
> |---|---|
> | No adversarial views generation         | 91.4 |
> |Random adversarial views         | 92.3 |
> | Ours (gradient-ascent guided adversarial views)         |**94.3**  |
>
> As presented in this table, our method outperforms the other two methods by at least 2\%, which proves the effectiveness of our proposed adversarial view generation method.
>
>
> **Q3:** The presentation is lacking clarity and details.
>
> **A3:** To our best, we have carefully revised the paper and corrected all grammar and writing issues for a better and clearer presentation. Moreover, we have added the missing reference in our revised manuscript according to the comments by the reviewer
>
> ```
> [1] Yongming Rao, Jiwen Lu, and Jie Zhou. Global-local bidirectional reasoning for unsupervised representation learning of 3d point clouds. In Proceedings of the IEEE Conference on Computer Vision and Pattern Recognition (CVPR), 2020.
>
> [2] Yi Shi, Mengchen Xu, Shuaihang Yuan, and Yi Fang. Unsupervised deep shape descriptor with point distribution learning. In Proceedings of the IEEE/CVF Conference on Computer Vision and Pattern Recognition, pp. 9353–9362, 2020
> ```

---

### Official Review · Reviewer_s4iu · 2021-11-03

**Correctness:** 4
**Technical Novelty And Significance:** 3
**Empirical Novelty And Significance:** 3
**Recommendation:** 6
**Confidence:** 4

**Main Review:**

- I find the main method of the paper interesting and practical. I particularly like the idea of identifying and rendering challenging viewpoints on the fly, compared to the more static setting of using a pre-rendered and pre-specified set of views adopted by many other works. The combination with contrastive learning is useful to avoid the extra requirement of class labels.

- Regarding rendering new views on the fly, I was wondering if there is a significant computational overhead when it comes to rendering these views, since this happens online during training? Although I like the idea, I can see how this can make training slow.

- In fact, how does this approach compare with just performing extreme data augmentations without mining hard negatives? I think this could be an interesting setting to check for the ablation study. In general, I think, that there are more choices that could be ablated in the main ablation study, instead of just the number of views (e.g., potentially a different hard negative mining method?).

- I like the presentation of the paper overall (e.g., the paper is clearly written, Figure 2 gives intuition about the type of hard examples that the method discovers), but there are quite a few typos (I list some of these below), so I would encourage the authors to do a more thorough proofread to eliminate those.
pg 5 : some wrong pointers leading to ??
pg 5 : "we can reformulate Equation" -> equation number missing
pg 6 : "a PyTorch3D" -> delete a
pg 6 : one-the-fly -> on-the-fly
pg 7 : we random rotate -> we randomly rotate
pg 7 : mash -> mesh
pg 7 : feces -> faces
pg 7 : Our MVCNN -> Our shouldn't be capitalized
pg 8 : "In this experiment, test the rotation" -> In this experiment, we test the rotation

- Evaluation happens on relatively saturated benchmarks, but quantitative performance is solid.

- The citations for the methods in Table 1 are missing. It would be much easier for the readers if the citations were also included in the Table, instead of only the names of the methods.

**Summary Of The Paper:**

This paper focuses on the problem of learning a descriptor for 3D shape. The two main contributions of the paper are related to using a mechanism for detecting hard examples for training the descriptor, while also doing it without requiring labels for this procedure. This is done with a combination from techniques inspired by adversarial learning and contrastive learning. The authors compare with a number of methods on the ModelNet40 benchmark.

**Summary Of The Review:**

All in all, I think this is an interesting paper, and the method could be applied in other settings as well. In fact, I would like to hear from the authors if they have other ideas for future extensions, because ModelNet is more of a simpler toy setting. I'm currently giving a Weak Accept rating, but I'm very interested in reading the response of the authors and the other reviews.

---

> ### Author Response · Authors · 2021-11-23
> **Responses to Reviewer s4iu**
>
> **Q1:** Regarding rendering new views on the fly, I was wondering if there is significant computational overhead when it comes to rendering these views since this happens online during training? Although I like the idea, I can see how this can make training slow.
>
>
> **A1:** To provide an accurate answer to the computational overhead, we conducted an additional experiment as follows. We first generated a collection of 2D view images offline through rending the 3D objects sampled from ModelNet40. Then we compared the computational overhead for the training with the offline rendered view images and for the training with online view generation (our method). We report the time of a forward pass for different methods in the table for one input sample during the training.
>
> |Method|Time|
> |---|---|
> |MVCNN  |39.89ms|
> |GVCNN        |40.12ms|
> |Ours(MV.) |43.39ms|
> |Ours(GV.) |43.62ms|
>
> In this table, the Ours(MV.) and Ours(GV.)  represent the MVCNN and GVCNN two different backbone networks that we used respectively.  As we can see from the experimental results, our online adversarial views generation during the training does not introduce a significant computational overhead. Please note the time reported in this table for MVCNN and GVCNN exclude the time cost by the offline rendering process.
>
> **Q2:** In fact, how does this approach compare with just performing extreme data augmentations without mining hard negatives? I think this could be an interesting setting to check for the ablation study. In general, I think, that there are more choices that could be ablated in the main ablation study, instead of just the number of views (e.g., potentially a different hard negative mining method?).
>
>
> **A2:** Thank you for the comments. Actually, in the supplementary material of our original submission, we included the related experimental results about the comparison with other data augmentation methods. The comparison results can be found in Table 5 of the supplementary submission. In addition, we further conduct an experiment by applying random rotation (as opposed to the gradient-ascent guided rotation) to input 3D instances to generate random adversarial views to verify the effectiveness of our proposed adversarial views. Specifically, we modify Equation 7 as:
> $$ \hat{\alpha},\hat{\beta},\hat{\gamma}= (p(\Delta\alpha,\Delta\beta,\Delta\gamma)+[\alpha,\beta,\gamma]),$$
> and we follow
> $$
> x^{adv}=
> \begin{cases}
> Mx& \text{i=0} \\\\
> \hat{M}x^{adv}_{i-1}& \text{i>0}
> \end{cases}
> $$
> to generate views by using random $\Delta\alpha,\Delta\beta,\Delta\gamma$ for each step of adversarial attack, where $M$ is a random initialized rotation matrix, and $\hat{M}$ is a rotation matrix controlled by $\hat{\alpha},\hat{\beta},\hat{\gamma}$. We use GVCNN as the backbone network and follow the same experimental settings in our main paper to train the new model. We report the classification accuracy on ModelNet40, denoted as "Random adversarial views''. Moreover, we compare our model with the one without using adversarial views. Specifically, we replace the adversarial view generation from our method with a random rotation operation (a single step), denoted as "No adversarial views generation'', to validate its effectiveness.
>
> |  Method  | Classification Acc.|
> |---|---|
> | No adversarial views generation         | 91.4 |
> |Random adversarial views         | 92.3 |
> | Ours (gradient-ascent guided adversarial views)         |**94.3**  |
>
> As presented in this table, our method outperforms the other two methods by at least 2\%, which proves the effectiveness of our proposed adversarial view generation method.
>
> **Q3:** The citations for the methods in Table 1 are missing. It would be much easier for the readers if the citations were also included in the Table, instead of only the names of the methods.
>
> **A3:** We have included the citations in the revised version.

---

> > ### Comment · Reviewer_s4iu · 2021-11-30
> > **Response to the authors' rebuttal**
> >
> > I would like to thank the authors for their response. They have addressed most of my questions and comments, particularly when it comes to comparing with other baseline augmentation schemes, and the overhead of online view generation. I am satisfied with the rebuttal and I lean towards keeping my original Weak Accept rating.

---

> > > ### Author Response · Authors · 2021-11-30
> > > **Re: Response to the authors' rebuttal (ICLR 2022 Conference Paper3813 Reviewer s4iu)**
> > >
> > > Thank you so much for your encouraging comments.

---

### Author Response · Authors · 2021-11-23
**General Responses to All Reviewers**

We sincerely thank reviewers for the detailed and constructive feedback. We appreciate the positive comments such as "an interesting and practical research problem'', "a particularly interesting idea'', "solid quantitative performance on saturated benchmarks'' (Reviewer: s4IU), "important and interesting contributions'' (Reviewer: 6V1K), "particularly interesting and novel method for adversarial views generation with good evaluations'' (Reviewer: GNMF), "an effective viewpoints selection method'' (Reviewer: aAcK). To our best, we have revised our paper to fully address the suggested weaknesses by adding suggested experiments, improving the presentation, and adding missing references (please refer to the revised manuscript for the details). We first summarize the common comments raised by reviewers and provide our responses to all reviewers below:

**Q1:** All reviewers raised questions regarding the effectiveness of the proposed method, and reviewers would like to see the comparison of our proposed adversarial view-based method with other data augmentation methods ( i.e., data augmentation with random rotations) to demonstrate the effectiveness of the proposed adversarial view.

**A1:** Thank you for the comments. Actually, in the supplementary material of our original submission, we included the related experimental results about the comparison with other data augmentation methods. The comparison results can be found in Table 5 of the supplementary submission. In addition, we further conduct an experiment by applying random rotation (as opposed to the gradient-ascent guided rotation) to input 3D instances to generate random adversarial views to verify the effectiveness of our proposed adversarial views. Specifically, we modify Equation 7 as:
$$ \hat{\alpha},\hat{\beta},\hat{\gamma}= (p(\Delta\alpha,\Delta\beta,\Delta\gamma)+[\alpha,\beta,\gamma]),$$
and we follow
$$
x^{adv}=
\begin{cases}
Mx& \text{i=0} \\\\
\hat{M}x^{adv}_{i-1}& \text{i>0}
\end{cases}
$$
to generate views by using random $\Delta\alpha,\Delta\beta,\Delta\gamma$ for each step of adversarial attack, where $M$ is a random initialized rotation matrix, and $\hat{M}$ is a rotation matrix controlled by $\hat{\alpha},\hat{\beta},\hat{\gamma}$. We use GVCNN as the backbone network and follow the same experimental settings in our main paper to train the new model. We report the classification accuracy on ModelNet40, denoted as "Random adversarial views''. Moreover, we compare our model with the one without using adversarial views. Specifically, we replace the adversarial view generation from our method with a random rotation operation (a single step), denoted as "No adversarial views generation'', to validate its effectiveness.

|  Method  | Classification Acc.|
|---|---|
| No adversarial views generation         | 91.4 |
|Random adversarial views         | 92.3 |
| Ours (gradient-ascent guided adversarial views)         |**94.3**  |

As presented in this table, our method outperforms the other two methods by at least 2\%, which proves the effectiveness of our proposed adversarial view generation method.

**Q2:** Writing clarity.

**A2:** To our best, we have carefully revised the paper and corrected all grammar and writing issues for a better and clearer presentation according to the comments by reviewers.

---

### Decision · Program_Chairs · 2022-01-20

**Decision:**

Reject

**Comment:**

This submission received a diverging set of the final ratings: 6, 3, 6, 5. On the positive side, reviewers appreciated practicality of the approach and supporting empirical results. At the same time, all of them expressed concerns with the presentation (typos, unfinished sentences, inconsistent notations). Additional requests for clarifications and ablation studies have been mainly addressed in the rebuttal. The most skeptical reviewer did not participate in the post-rebuttal discussion, thus the final decision took that into account..

The AC has read the paper and verified that the minor technical issues pointed out by the reviewers have been fixed in the updated version (there are still a couple of typos remaining). This submission was further discussed between AC and SAC, as well as in the PC calibration meeting. Both AC and SAC agreed with the comment of Reviewer aAcK who pointed out that generating adversarial samples for mining hard examples has been explored in more general but related contexts before, which limits the novelty of this work to an application of a known idea to a particular domain (3D). At the same time, performance gains on the ModelNet40 dataset are marginal compared to the point cloud based baselines, while the proposed method still uses point clouds for generating adversarial views. In combination with other minor issues pointed out by the reviewers, and given that none of the reviewers was championing the paper, AC and SAC believe that the weaknesses of this paper at the end outweigh its strengths and do not recommend acceptance at this stage.